# DEEP REINFORCEMENT LEARNING WITH CAUSALITY-BASED INTRINSIC REWARD

## ABSTRACT

Reinforcement Learning (RL) has shown great potential to deal with sequential decision-making problems. However, most RL algorithms do not explicitly consider the relations between entities in the environment. This makes the policy learning suffer from the problems of efficiency, effectivity and interpretability. In this paper, we propose a novel deep reinforcement learning algorithm, which firstly learns the causal structure of the environment and then leverages the learned causal information to assist policy learning. The proposed algorithm learns a graph to encode the environmental structure by calculating Average Causal Effect (ACE) between different categories of entities, and an intrinsic reward is given to encourage the agent to interact more with entities belonging to top-ranked categories, which significantly boosts policy learning. Several experiments are conducted on a number of simulation environments to demonstrate the effectiveness and better interpretability of our proposed method.

## 1 INTRODUCTION

Reinforcement learning (RL) is a powerful approach towards dealing with sequential decision-making problems. Combined with deep neural networks, deep reinforcement learning (DRL) has been applied in a variety of fields such as playing video games (Mnih et al., 2015; Vinyals et al., 2019; Berner et al., 2019), mastering the game of Go (Silver et al., 2016) and robotic control (Riedmiller et al., 2018). However, current DRL algorithms usually learn a black box policy approximated by a deep neural network directly using the state transitions and reward signals, without explicitly understanding the structure information of the environment.

Compared with DRL agents, an important reason why humans are believed to be better at learning is the ability to build model on the relations between entities in the environment and then reason based on it. This ability is an important component of human cognition (Spelke & Kinzler, 2007). As the learning process continues, through interactions with the environment and observations of it, human can gradually understand its actions' causal effects on the entities as well as the relations between entities and then reason based on them to figure it out the most important actions to take in order to improve the efficiency. In scenarios that contain multiple entities with complicated relations, optimal policy may be obtained only when the structured relation information is captured and exploited. However, most current DRL algorithms do not consider structured relation information explicitly. The knowledge learned by an agent is implicitly entailed in the policy or action-value function, which are usually unexplainable neural networks. Therefore, whether the relations are well understood and exploited by the agent is unknown. When the environment is with high complexity, blackbox learning of policies suffers from low efficiency, while policy learning over explicit representation of entity relations can significantly boost the learning efficiency. Based on the fact that entities in an environment are often not independent but causally related, we argue that disentangling the learning task into two sequential tasks, namely relational structure learning and policy learning, and leveraging an explicit environmental structure model to facilitate the policy learning process of DRL agents are expected to boost the performance. With the learned relational structure information, the agent performs exploration with a tendency of prioritizing interaction with critical entities, which is encouraged by intrinsic rewards, to learn optimal policy effectively.

Taking this inspiration, we propose a deep reinforcement learning algorithm which firstly learns the relations between entities and then recognize critical entity categories and develop an intrinsic reward based approach to improve policy learning efficiency and explainability. The proposed algo-

rithm learns a graph to encode the relation information between categories of entities, by evaluating causal effect of one category of entities to another. Thereafter, intrinsic reward based on the learned graph is given to an agent to encourage it to prioritize interaction with entities belonging to important categories (the categories that are root causes in the graph). Previous works also use graphs to provide additional structured information for the agent to assist policy learning (Wang et al., 2018; Vijay et al., 2019). However, graphs leveraged by these works are provided by human and thus rely heavily on prior knowledge. Compared with their methods, our algorithm overcomes the deficiency that the graph can not be generated automatically. Our approach requires no prior knowledge and can be combined with existing policy-based or value-based DRL algorithms to boost their learning performance. The key contributions of this work are summarized as follows:

- We propose a novel causal RL framework that decomposes the whole task into the structure learning and causal structure aware policy learning.
- The learned causal information is leveraged by giving causality based intrinsic reward to an agent, to encourage it to interact with entities belonging to critical categories for accomplishing the task.
- We design two new game tasks which contain multiple entities with causal relations as benchmarks to be released to the community. The new benchmarks are designed in such ways that categories of objects are causally related. Experiments are conducted on our designed simulation environments, which show that our algorithm achieves state-of-the-art performance and can facilitate the learning process of DRL agents under other algorithmic frameworks.

The paper is organized as follows. In Section 2, we introduce deep reinforcement learning and Average Causal Effect (ACE), which are key components of this work. Then we illustrate our algorithm in Section 3 in details. In Section 4, we show the experimental results on the designed environment to demonstrate the effectiveness of our framework. In Section 5, we introduce previous works that relate to our method. Finally, conclusions and future work are provided in Section 6.

## 2 BACKGROUND

### 2.1 DEEP REINFORCEMENT LEARNING

An MDP can be defined by a 5-tuple $(\mathcal{S}, \mathcal{A}, \mathcal{P}, \mathcal{R}, \gamma)$, where $\mathcal{S}$ is the state space, $\mathcal{A}$ is the action space, $\mathcal{P}$ is the transition function, $\mathcal{R}$ is the reward function and $\gamma$ is the discount factor (Sutton & Barto, 2018). A RL agent observes a state $s_t \in \mathcal{S}$ at time step $t$. Then it selects an action $a_t$ from the action space $\mathcal{A}$ following a policy $\pi(a_t|s_t)$, which is a mapping of state space to action space. After taking the action, the agent receives a scalar reward $r_t$ according to $\mathcal{R}(s_t, a_t)$. Then the agent transits to the next state $s_{t+1}$ according to the state transition probability $\mathcal{P}(s_{t+1}|s_t, a_t)$. A RL agent aims to learn a policy that maximizes the cumulative discount reward, which can be formulated as $R_t = \sum_{k=0}^{T} \gamma^k r_{t+k}$ where $T$ is the length of the whole episode. In the process of learning an optimal policy, a RL agent generally approximates the state-value function $V_\pi(s)$ or the action value function $Q_\pi(s, a)$. The state value function is the expected cumulative future discounted reward from a state with actions sampled from a policy $\pi$:

$$V_\pi(s) = \mathbb{E}_\pi \left[ \sum_{k=0}^{T} \gamma^k r_{t+k} | S_t = s \right]. \tag{1}$$

Deep Reinforcement Learning (DRL) which combines Deep Neural Networks (DNNs) with RL can be an effective way to deal with high-dimensional state space. It benefits from the representation ability of DNNs, which enable automatic feature engineering and end-to-end learning through gradient descent.

Several effective algorithms have been proposed in the literature and we use A2C in this paper as our basic algorithm, which is a synchronous version of A3C (Mnih et al., 2016). A2C consists of a variety of actor-critic algorithms (Sutton et al., 2000). It directly optimizes the policy $\pi_\theta$ parameterized by $\theta$ to maximize the objective $J(\theta) = \mathbb{E}_\pi[\sum_{k=0}^{T} \gamma^k r_{t+k}]$ by taking steps in the direction of $\nabla_\theta J(\theta)$. The gradient of the policy can be written as:

$$\nabla_\theta J(\theta) = \mathbb{E}_\pi[\nabla_\theta \log \pi_\theta(a|s) A^\pi(s, a)], \tag{2}$$

where $A^\pi(s, a) = Q^\pi(s, a) - V^\pi(s)$ is the advantage function. The advantage function can be estimated by one-step TD-error $\hat{A}^\pi(s_t, a_t) = r_t + \gamma V_\phi(s_{t+1}) - V_\phi(s_t)$, where $V_\phi(s)$ is the approximation of the state-value function $V^\pi(s)$ parameterized by $\phi$.

## 2.2 CAUSAL NEURAL NETWORK ATTRIBUTIONS

Attributions are defined as the effect of an input feature on the prediction function's output (Sundararajan et al., 2017) and Chattopadhyay et al. (2019) propose a neural network attribution methodology built from first principles of causality. Chattopadhyay et al. (2019) views the neural network as a Structural Causal Model (SCM), and proposes a new method to compute the Average Causal Effect of an input neuron on an output neuron based on the $do(.)$ calculus (Pearl, 2009).

**Definition 1.** *(Average Causal Effect). The Average Causal Effect (ACE) of a binary random variable $x$ on another random variable $y$ is commonly defined as $\mathbb{E}[y|do(x = 1)] - \mathbb{E}[y|do(x = 0)]$.*

While the above definition is for binary-valued random variables, the domain of the function learnt by neural networks is usually continuous. Given a neural network with input layer $l_1$ and output layer $l_n$, we hence measure the ACE of an input feature $x_i \in l_1$ with value $\alpha$ on an output feature $y \in l_n$ as:

$$ACE^y_{do(x_i=\alpha)} = \mathbb{E}[y|do(x_i = \alpha)] - baseline_{x_i}. \tag{3}$$

**Definition 2.** *(Causal Attribution). The causal attribution of input neuron $x_i$ for an output neuron $y$ is defined as $ACE^y_{do(x_i=\alpha)}$.*

In Equation 3, an ideal baseline would be any point along the decision boundary of the neural network, where predictions are neutral. However, Kindermans et al. (2019) showed that when a reference baseline is fixed to a specific value (such as a zero vector), attribution methods are not affine invariant. Therefore, Chattopadhyay et al. (2019) proposed the average ACE of $x_i$ on $y$ as the baseline value for $x_i$:

$$baseline_{x_i} = \mathbb{E}_{x_i}[\mathbb{E}_y[y|do(x_i = \alpha)]] \tag{4}$$

In this paper, we use causal attribution method to infer the relations between entities.

## 3 METHOD

In this section, we present a novel DRL framework named CARE (**CA**usal **RE**lation) that enables the agent to infer the causal relationship between entities. Figure 1 illustrates the overall framework of CARE. CARE adopts a two-stage training paradigm. In the first stage, the agent learns a model of the environment by minimizing the prediction error of the state transition. Then, we calculate ACE values between categories of entities, which are used for constructing a causal graph $\mathcal{G}$. When $\mathcal{G}$ is at hand, we are able to obtain the causal ordering of all categories, that is, a permutation such that nodes ranked lower cannot cause nodes ranked higher. This order is used to measure the importance of the categories, and intrinsic reward is given based on the causal ordering. Specifically, the agent receive a one-step "enhanced" reward $r_t$ where:

$$r_t = r_t^{\mathcal{G}} + r_t^{ext}, \tag{5}$$

where $r_t^{ext}$ is the original extrinsic reward given by the environment, and $r_t^{\mathcal{G}}$ is the intrinsic reward, designed by the learning algorithm to encourage the agent to maximize the effect of its behaviour on the change of states of critical entities. We describe the details in later sections.

This section is organized as follow. In section 3.1, we first introduce category-oriented state factorization. In section 3.2, we describe how to get the relation graph $\mathcal{G}$ and in section 3.3, we show how to calculate the intrinsic reward $r_t^{\mathcal{G}}$.

## 3.1 CATEGORY-ORIENTED STATE FACTORIZATION

We focus on environments which contain multiple categories of entities. The category of entity is the acting rules that govern the actions of the entity. Each entity is within one category, and the entities within one category share the same acting rules. An example of the category of entity appears in the experimental section. Consider an environment consisting of two kinds of sheep:

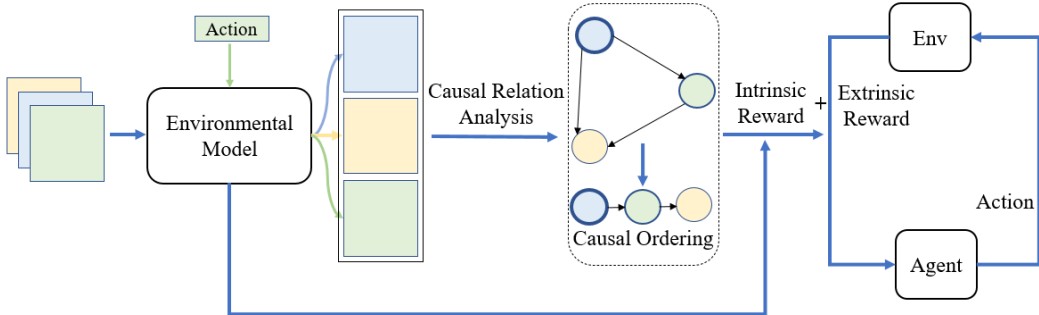

Figure 1: The Overall framework of CARE. CARE firstly learns a model of the environment which takes the states of different categories of entities as input and predicts the next step states of all categories. Then we perform causal relation analysis on the model to get a relation graph among all categories. Finally, intrinsic reward based on the graph and the effect of agent behavior on entities is given to the agent to assist policy learning.

one is the ewe that takes random walk. The other is the lamb that always follow the ewe. Then there are two categories of entities in the environment. In this paper, we infer the causal relations among categories. Another view is to infer the causal graph among all entities in the environment. However, certain abstraction of entities is beneficial and simplifies the learning because quite often in a dynamic and interactive environment, entities could pop up or disappear as the result of actions taken by the agent or the environmental evolution. Therefore, maintaining a graph with changing nodes could be quite challenging, rendering the learning algorithm unnecessarily complicated. We thus choose category-level casuality inference for scalability. The category of each entity is given as a prior, or generated by applying computer machine vision technology such as unsupervised or pretrained object segmentation models (Agnew & Domingos, 2020), shape and color analysis (Ren et al., 2015; He et al., 2017). We introduce a factored space to represent the states. The factored state space $\mathcal{S}$ consisting of entities of $K$ categories is $\mathcal{S} = \mathcal{S}^1 \times ... \times \mathcal{S}^K$, where $\mathcal{S}^i$ is the state space of the $i$th category. At time $t$, the state of the entities of $i$th category is $s_t^i \in \mathcal{S}^i$, and the state $s_t \in \mathcal{S}$ of all entities is composed of local states of all category $s_t = [s_t^1, s_t^2, ..., s_t^K]$. This factorization ensures that each category of entities is independently represented. In this paper, the state is represented using a $K$-channel feature map, each corresponding to one category of entities. More details can be found in the Appendix.

### 3.2 CAUSAL RELATION ANALYSIS

In this section, we demonstrate how to obtain the casual graph. Firstly, we learn a model of the environment, which predicts the next step state of each category of entities. Thereafter, we perform average causal effect analysis between each pair of categories. Namely, conditioning on all other categories, we compute a measurement quantifying the influence of one category on the other. Based on this, we are able to recover the whole causal graph, and the causal ordering of categories. In hypothesis, vertices with higher ranking are more important in the environment, and we will give intrinsic rewards based on the influence of the agent's action on different categories of entities.

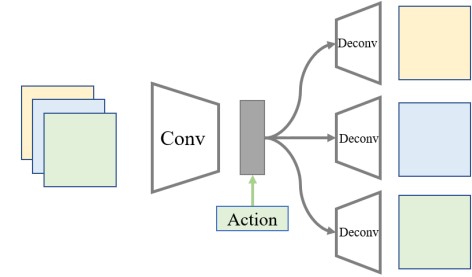

Figure 2: Architecture of the environmental model. The model contains one encoder and $K$ decoders each corresponding to one category. The encoder takes current state as input and the decoders predict the next step state of each category.

**Environment Model Learning**

To learn the environment model, we first use a random agent to interact with the environment to collect a buffer of experience $\mathcal{B} = \{(s_t, a_t, s_{t+1})\}_{t=1}^T$.

It contains $T$ tuples of states $s_t \in \mathcal{S}$, actions $a_t \in \mathcal{A}$, and follow-up states $s_{t+1} \in \mathcal{S}$, which is reached after taking action $a_t$. Our goal is to predict next step state of each category of entities to understand the environment, without agent's policy training. It should be noted that our model is only used to analyze the relations between category of entities, i.e. the causal ordering of all entity categories, instead of using the model for planning like model-based RL. Therefore, our method does not require extremely high model accuracy.

Our model employs an encoder-decoder architecture (See Figure 2). The encoder is a CNN operating directly on observations from the environment, and the output of the encoder $z_t = Enc(s_t, a_t)$ is flattened to a vector. The decoder is composed of several deconvolutional layers. There are $K$ decoders $Dec^1...Dec^K$, each corresponding to a category of entities, which take the encoded vector as input and predict the next state of each category of entities.

$$\hat{s}_{t+1}^k = Dec^k(z_t) \tag{6}$$

The model is trained by minimizing the following loss:

$$\mathcal{L} = \sum_{k=1}^{K} d(s_{t+1}^k, \hat{s}_{t+1}^k) \tag{7}$$

where $d(s_{t+1}^k, \hat{s}_{t+1}^k)$ denotes the distance between the predicted next state $\hat{s}_{t+1}^k$ and the true one.

**Calculating ACE**

After getting the environment model trained, we can calculate ACE values of each pair of categories following the method described in Section 2.2, which computes ACE by intervening the states of each category of entities. Specifically, the ACE of a category $i$ on another category $j$ can be calculated by:

$$ACE_{do(s_t^i=s_\tau^i)}^{s_{t+1}^j} = \mathbb{E}[s_{t+1}^j|do(s_t^i = s_\tau^i)] - baseline_{s_t^i} \tag{8}$$

Here $s_\tau^i \in \mathcal{S}^i$ is the interventional state. $do(s_t^i = s_\tau^i)$ means that we manually control $s_t^i$ to another state $s_\tau^i$, known as $do$ calculus in causal analysis. The $baseline_{s_t^i}$ is calculated by:

$$baseline_{s_t^i} = \mathbb{E}_{s_t^i}[\mathbb{E}_{s_{t+1}^j}[s_{t+1}^j|do(s_t^i = s_\tau^i)]] \tag{9}$$

By definition, the interventional expectation $\mathbb{E}[s_{t+1}^j|do(s_t^i = s_\tau^i)]$ is written as

$$\mathbb{E}[s_{t+1}^j|do(s_t^i = s_\tau^i)] = \int s_{t+1}^j p(s_{t+1}^j|do(s_t^i = s_\tau^i)) \, ds_{t+1}^j \tag{10}$$

Computing the integral is intractable because the exact distribution of $s_{t+1}^j$ is unknown. Thus, we approximate Equation 10 by empirical historical distribution sampling:

$$\mathbb{E}[s_{t+1}^j|do(s_t^i = s_\tau^i)] \approx \frac{1}{N} \sum_{(s_m, a_m, s_{m+1}) \in \mathcal{B}_N} \hat{s}_{m+1}^j \tag{11}$$

where $\hat{s}_{m+1}^j = Dec^j(Enc(\dot{s}_m^{(i)}, a_m))$ is the predicted next state of category $j$ and $\dot{s}_m^{(i)} = [s_m^1, ..., s_\tau^i, ..., s_m^K]$ is the interventional state which sets $s_m^i = s_\tau^i$ while leaves other categories unchanged. $\mathcal{B}_N \subseteq \mathcal{B}$ is a batch of experience sampled from the buffer with sample size $N$. The maximal ACE value is used as the final effect of category $i$ on category $j$:

$$ACE_{i \to j} = \max_{s_\tau^i \in \mathcal{S}^i} (ACE_{do(s_t^i=s_\tau^i)}^{s_{t+1}^j}) \tag{12}$$

In practice, it is also computed by sampling from the historical states set of the category $i$.

After getting pairwise ACE values, we are able to get the causal graph $\mathcal{G} = (\mathcal{V}, \mathcal{E})$ of all categories of entities. $\mathcal{V}$ is the set of all vertices and each vertex represents a category of entities. $\mathcal{E}$ is the set of all edges and $e_{ij} \in \mathcal{E}$ represents that category $i$ causes category $j$. Let $H$ be the $K \times K$ adjacency matrix of $\mathcal{G}$, which is obtained by the edge directing rule:

$$H_{ij} = \begin{cases} 1, & \text{if } ACE_{i \to j} > ACE_{j \to i} \\ 0, & \text{otherwise} \end{cases} \tag{13}$$

Since $\mathcal{G}$ is assumed to be a directed acyclic graph (DAG), there are no feedback loops or any path starting from category $i$ and then back to itself. Consequently, there exists a causal ordering of all vertices. Causal ordering is a permutation $\mu$ of all index of vertices $\{1, ..., K\}$, where vertices ranked higher cannot be caused by ones ranked lower. The nodes ranked higher are hypothetically more critical entity categories for the task. This will be used for designing intrinsic reward. Based on this, we define the criticality of entity category.

**Definition 3.** *(Criticality of Entity Category). The criticality of entity category is defined as the ranking of the category in the causal ordering $\mu$.*

### 3.3 Intrinsic Reward

We adopt the tendency to prioritize critical entities by giving intrinsic rewards to the agent in addition to the original extrinsic reward for policy learning. The basic idea is that actions that have a relatively large effect on entities whose category ranks higher in the causal ordering are rewarded. Based on the learned model as described in Section 3.2, we define the effect $I_i(s_t, a_t)$ of the agent's behavior on the $i$th category of entities, similar to ACE as:

$$I_i(s_t, a_t) = f_i(s_t, a_t) - \frac{1}{|\mathcal{A}|} \sum_{a \in \mathcal{A}} f_i(s_t, a) \tag{14}$$

Here $f_i(s_t, a_t) = Dec^i(Enc(s_t, a_t))$ denotes the learned model and $|\mathcal{A}|$ is the size of the action space. This method calculates the effect of the agent's action on the a certain category of entities and the second term $\frac{1}{|\mathcal{A}|} \sum_{a \in \mathcal{A}} f_i(s_t, a)$ in equation 14 serves as the *baseline* when calculating ACE.

The intrinsic reward is defined as $r_t^{\mathcal{G}} = \sum_{i=1}^{K} r_t^{\mathcal{G},i}$. For each category, the intrinsic reward is:

$$r_t^{\mathcal{G},i} = \begin{cases} \beta_i, & \text{if } I_i(s_t, a_t) > \delta \\ 0, & \text{otherwise} \end{cases} \tag{15}$$

Here $\beta_i$ and $\delta$ are hyperparameters. It is constrained that the rewards of categories along the causal ordering are non-increasing $\beta_{\mu_1} \geq \beta_{\mu_2} \geq ... \geq \beta_{\mu_K}$, where $\mu_i$ corresponds to position $i$ in the causal ordering.

## 4 Experimental Results

In this section, we evaluate CARE on two test domains (see Figure 3), Shepherd and Ant, where an agent is required to have the ability of relational reasoning to learn an ideal policy. We compare CARE with flat DRL method (A2C) and Relational DRL (Zambaldi et al., 2018), which is an A2C based algorithm and uses multi-head attention to extract relation between entities. Experiment details can be found in the Appendix.

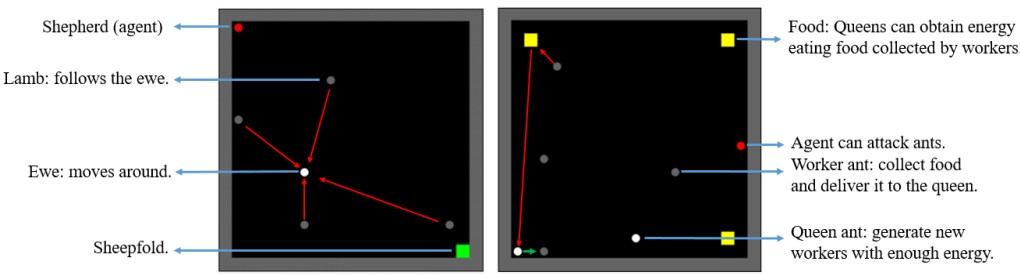

Figure 3: The Shepherd game ($left$) and the Ant game ($right$).

### 4.1 Shepherd

In the Shepherd game (Figure 3, $left$), a shepherd is expected to drive all of the ewe and lambs, which are randomly distributed in the ranch at the beginning, back into the sheepfold. The agent's

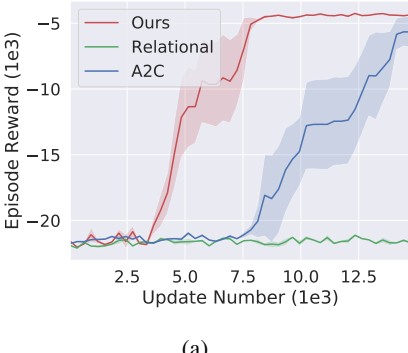 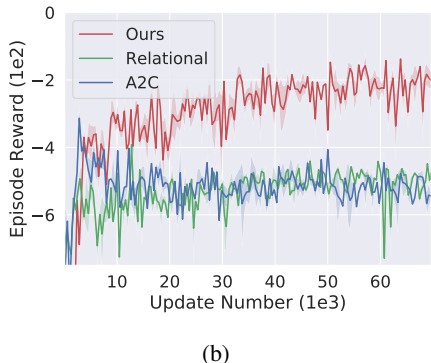

(a)               (b)

Figure 4: Experimental results on the two test games.$(a)$ Learning curves of the Shepherd game. $(b)$ Learning curves of the Ant game. All experiments are averaged over 5 random seeds.

objective is to finish the task using as little time as possible. Unless the shepherd gets close and drives the sheep, the ewe will walk around the ranch and the lambs follow the ewe. The shepherd is viewed as the agent here and it has five actions: $up, down, left, right$ and $drive$. Each action makes the agent move to the corresponding direction with one step size except for $drive$. If the agent takes the $drive$ action, sheep near the agent will be driven to move towards the sheepfold. The game ends when all sheep are in the sheepfold or the time exceeds a fixed period. At every step, the agent receives a reward equal to the negative sum of distance from each sheep to the sheepfold. Here we have $K = 2$ categories of entities.

We firstly evaluate our algorithm in this game. The experimental result is shown in Figure 4(a). The result shows that our method converges to a better policy with higher mean episode return than other methods. Flat A2C agent can also learn a relatively good policy with slightly worse performance compared with our method, but it takes a longer time. This result shows that understanding the causal relations and leveraging the learned relational information can significantly boost the learning process of DRL agents. The performance of the Relational DRL algorithm is not to our expectation, possibly because the attention mechanism does not capture correct relations of the entities in the environment.

The causal graph learned by CARE is in Figure 6 ($left$). Here $E$, $L$ and $A$ denote $Ewe$, $Lamb$ and $Agent$ respectively. The edge from $Ewe$ to $Lamb$ represents that lambs are attracted by the ewe. The two edges from the $Agent$ to $Ewe$ and $Lamb$ means that the ewe and lambs are driven by the agent. Although the agent is also considered a node, it is not ranked in the causal ordering. The ACE values for getting this graph are listed in Table 1.

We also evaluate the effect $I_i(s, a)$ calculated by Equation 14, which is the cornerstone of the intrinsic reward. We firstly sampled a state from the historical trajectories, and then manually set the agent to every grid in the field. At each grid, we calculate the agent's effect on the category of $ewe$ using Equation 14. Finally we visualize the calculated effect in Figure 5. The value of coordinate $(x, y)$ in the heatmap corresponds to effect of the agent on the $ewe$ when it is on position $(x, y)$ and taking action $drive$. As shown in the Figure 5, the calculated $I_{ewe}(s, drive)$ is high only when the agent is near the ewe. This is be-

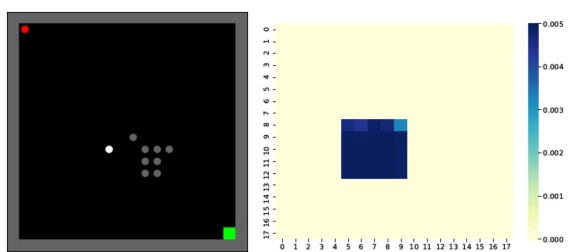

Figure 5: Visualization of the $I_{ewe}(s, drive)$ in the Shepherd game. The heatmap is generated by intervening the position of the agent to every grid in the field to get $I_{ewe}(s, drive)$ using Equation 14.

cause the $ewe$ will be driven to move towards the sheepfold only when the shepherd and sheep are close enough. This result shows that the effect of the agent on the target category of entities is well

modeled. Moreover, this result also shows that we can choose the hyperparameter $\delta$ easily, because there is a large gap between the calculated $I_i(s, a)$ when the agent's behavior affects $i$th category of entities or not.

Table 1: ACE values in the Shepherd game.

| $ACE$ | $ewe$ | $lamb$ | $agent$ |
|---|---|---|---|
| $ewe$ | – | **1.52e-03** | 1.03e-06 |
| $lamb$ | 5.14e-05 | – | 1.06e-06 |
| $agent$ | **3.65e-05** | **6.44e-06** | – |

Table 2: ACE values in the Ant game.

| $ACE$ | $queen$ | $worker$ | $food$ | $agent$ |
|---|---|---|---|---|
| $queen$ | – | **2.97e-03** | **7.71e-04** | 1.45e-06 |
| $worker$ | 4.33e-05 | – | 2.42e-03 | 4.13e-06 |
| $food$ | 1.22e-06 | **3.72e-03** | – | 2.72e-07 |
| $agent$ | **1.76e-06** | **2.12e-04** | **3.86e-05** | – |

## 4.2 ANT

In the Ant game (Figure 3, $right$), the agent is expected to kill all ants in the field. There are two queen ants and four worker ants at the beginning. Queen ants move around the field. Worker ants will firstly go to the nearest food and then bring it to a queen ant. The queen ant obtains some energy by eating the food. If the queen ant's energy exceeds a threshold, it will generate a new worker ant. Food will continue to be produced in the fixed positions. For this environment, there are $K = 3$ categories of entities. The agent has five actions: $up, down, left, right$, each of which makes the agent move one step towards the corresponding direction and $attack$, which kill the ant around the agent if there exists. The game ends when all ants are killed or the time span exceeds a fixed period. The agent will receive a reward of +1 if it kill one ant, whether it is a worker ant or a queen ant. At the end of the episode, the agent will receive a reward of $-(10 \times n + 100 \times m)$, where $n$ and $m$ denote the number of left worker ants and queen ants respectively.

An optimal policy should prioritize the task of killing the queen ant. Otherwise, worker ants will be continuously produced by the queen ants and the number of ants grows up very fast. We evaluate our algorithm in this game, comparing to flat A2C and Relational DRL. The experimental results are given in the Figure 4(b). In this game, we observe that our method learns a policy that kills the queen ants first and then other ants. However, flat A2C and Relational DRL both learn a policy that keeps the queen ants alive but stays at a certain position around the food to wait for and kill worker ants coming for food.

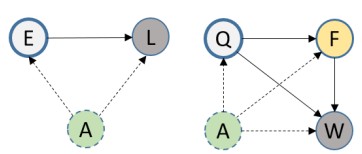

Figure 6: Learned graphs of the Shepherd game ($left$) and the Ant game ($right$).

We show the learned causal graph in Figure 6 ($right$). $Q, W, F$ and $A$ denote $Queen, Worker, Food$ and $Agent$ respectively. The causal ordering is $[Q, F, W]$ and the calculated ACE values can be found in the Table 2.

Since our algorithm has a graph learning procedure, we also record the running time of the algorithms in Table 3. Our algorithm takes longer time than A2C when the training step is the same. However, given the performance gain, we think the cost in time is reasonable: our algorithm learns a better policy, which is not obtainable by other algorithms. The performance gap is especially obvious in Ant game.

Table 3: Running time (seconds) in the Shepherd and Ant games.

| Method | Shepherd | | Ant | |
|---|---|---|---|---|
| | Graph learning | RL policy learning | Graph learning | RL policy learning |
| CARE | 1002 | 1149 | 3201 | 5136 |
| A2C | – | 779 | – | 4516 |
| Relational | – | 2251 | – | 9919 |

## 5 RELATED WORK

Our framework learns a graph and the graph entails the relationship between entities in the environment. Compared with model-based reinforcement learning algorithms (Sutton, 1991; Silver et al.,

2008; Heess et al., 2015; Nagabandi et al., 2018), which usually learn environment dynamics and plan or learn on the learned dynamics to reduce the interaction with the real environment, our method focuses on learning relations of entities but not environment dynamics. The learned causal graph is used to order the category of entity. Mahadevan & Maggioni (2007); Mabu et al. (2007); Metzen (2013); Shoeleh & Asadpour (2017); Sanchez-Gonzalez et al. (2018) also use graphs to learn a representation of the environment. However, these methods still focus on learning environment dynamics and thus these problems are usually solved via model-based RL.

The learned graph could be viewed as a structure description of the environment. Applying structure knowledge of environments in RL has been studied in previous works. Wang et al. (2018) explicitly models the structure of an agent as a graph and uses a GNN to approximate a policy. Vijay et al. (2019) builds a knowledge graph as prior for the agent, which illustrates different relations between entities in the environment. However, graphs leveraged by these two works are priors provided by human. Compared with these works, our algorithm supports automatic graph learning and requires no human prior knowledge. Ammanabrolu & Riedl (2019) proposed KG-DQN, which constructs a knowledge graph to represent the environment and uses Graph Neural Networks to extract features of the graph. This work nevertheless only adapts to Text-Adventure Game, because their knowledge graph can be only generated from natural language. Zambaldi et al. (2018) use multi-head attention to extract relation between entities. However, their method solves problem from the aspect of entity instead of category. Notice that our model deploys a encoder-decoder structure for processing the input signals. This is used by Shi et al. (2020) known as self-supervised interpretable network for extracting task-relevant attention masks which are interpretable features for agent's decisions. Agnew & Domingos (2020) uses object-centric state representations and exploits the object interactions and dynamics for identifying task-relevant object representations.

## 6 CONCLUSIONS

In this paper, we propose a novel deep reinforcement learning algorithm, which firstly learns environmental causal structure and then leverages the learned relational information to assist policy learning. Experimental results show that our algorithm has good performance, indicating that incorporating the environmental structure for reasoning is a promising research direction. Future work includes studying environments with dynamical graphs, and improving the training efficiency of the framework.

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

# A APPENDIX

## A.1 EXPERIMENTAL SETTINGS

**State Representation**

In both the Shepherd game and the Ant game, the whole game field is a grid world with size $18 \times 18$. The game state is represented by a tensor of size $K \times 18 \times 18$. Each channel represents the state of one category of entities.

**Environmental model**

The encoder is composed of three convolutional layers. The first, second and third layer have 64, 128 and 256 $3 \times 3$ filters with stride 2 respectively, each followed by a Batch Normalization layer. The output of the encoder is flatten to a vector and then enter two fully connected layers both with 512 hidden units. The action is encoded into a vector of size 32 by three FC layers with 100 hidden units and then is concatenated with the output of the encoder. Each decoder takes the output of encoder as input and pass it into four fully connected layers, with 900, 300, 512, 512 hidden units respectively. Then the output of the FC layers is taken as input of three deconvolutional layers. The first two layers have 128 and 64 $3 \times 3$ filters with stride 2 respectively. The last layer has $num\_categories$ $4 \times 4$ filters also with stride 2. Each deconvolutional layer is also followed by a Batch Normalization layer.

When training the model, we collect a buffer with 5000 episodes. We use the Adam optimizer to train the model with a learning rate of $1e - 4$ and a batch size of 32.

**Parameter Setting of RL**

CARE and flat A2C use the same network architecture. The actor and critic share the same first two convolutional layers and a FC layer. The two convolutional layers have 64 and 32 $3 \times 3$ filters with stride 2 respectively. The FC layer has 128 hidden units. The critic and the actor both take the output of the share part and pass it into two FC layers with 512 hidden units. Finally they output the value and the action distribution. For the Relational DRL algorithm, we use an open source implementation[1]. Other parameters are listed as follow:

---

[1]https://github.com/mavischer/DRRL

Table 4: Shepherd

| − | CARE | A2C | Relational |
|---|---|---|---|
| Number of process | 16 | 16 | 16 |
| Discount factor | 0.99 | 0.99 | 0.99 |
| Optimizer | RMSProp | RMSProp | RMSProp |
| Learning rate | $7e-4$ | $7e-4$ | $7e-4$ |
| Entropy term coefficient | 0.01 | 0.01 | 0.01 |
| $\delta$ | $1e-3$ | − | − |
| $\beta$s | $(10,0)$ (decays to 0 at 3000th update) | − | − |

Table 5: Ant

| − | CARE | A2C | Relational |
|---|---|---|---|
| Number of process | 16 | 16 | 16 |
| Discount factor | 0.99 | 0.99 | 0.99 |
| Optimizer | RMSProp | RMSProp | RMSProp |
| Learning rate | $7e-4$ | $7e-4$ | $7e-4$ |
| Entropy term coefficient | 0.5 | 0.1 | 0.1 |
| $\delta$ | $8e-4$ | − | − |
| $\beta$s | $(100,0,0)$ | − | − |

