# OpenReview forum: "Deep Reinforcement Learning with Causality-based Intrinsic Reward"
_ICLR.cc/2021/Conference — Reject_

### Official Review · AnonReviewer3 · 2020-10-28
**Marginally below acceptance threshold**

**Rating:** 5
**Confidence:** 3

**Review:**

Summary:
The paper proposes a deep reinforcement learning algorithm of advantage actor-critic (A2C), which ﬁrstly learns the causal structure of the environment and then leverages the learned causal information to assist policy learning. The causal structure is computed by calculating Average Causal Effect (ACE) between different categories of entities, and the authors assume that an intrinsic reward is given to encourage the agent to interact more with critical entities of the causal graph. Two experiments were conducted on simulation environments (Shepherd and Ant tasks) and demonstrated the effectiveness in obtaining the rewards and interpretable and accurate detection of the true graph.

Reasons for score:
The motivation and solutions were simple and clear, but the related work about deep reinforcement learning leveraging causal relation was not properly introduced. Furthermore,  there were unclear points in the implementation (without sharing codes) and the results. I think the idea is interesting and contributed to this community, but for the above reasons, it is difficult to provide a higher rating.


Pros:
1. The authors designed causality-based intrinsic reward to an agent, to encourage it to interact with critical entities for accomplishing the task.
2. Two experiments were conducted on simulation environments (Shepherd and Ant tasks with heterogeneous agents) and demonstrated the effectiveness in obtaining the rewards and interpretable and accurate detection of the true graph.

Cons:
1. There were also other researches in deep reinforcement learning leveraging causal relation (e.g., [i] and [ii] below). The title and the first contribution of this paper should be more specific and the related work should be included in the main text.
2. There were unclear points in the implementation (without sharing codes) and the results (below).

[i] W. Shi et al.  Self-Supervised Discovering of Causal Features: Towards Interpretable Reinforcement Learning, arXiv, 2020
[ii] W. Agnew and P. Domingos. Relevance-Guided Modeling of Object Dynamics for Reinforcement Learning, arXiv, 2020

Other comments:

There is little information about the algorithm of Relational DRL (Zambaldi et al., 2018) as one of the baseline. Relational DRL seems to be an attention-based A2C algorithm, but is the difference only related to the attention-based approach? At least, whether it is an A2C algorithm or not should be clarified.

In both experiments, is the number of categories K the number of agents? This was not specified.

I did not understand Figure 5 right and the sentence (”The value of coordinate (x, y) in the heatmap corresponds to the effect of the agent on the ewe when it is on position (x, y) and taking action drive”). The correspondence of xy coordinates with Figure 5 left might help my understanding.

---

> ### Author Response · Authors · 2020-11-17
> **Thank you for your valuable comments. Here are our responses.**
>
> **Q: “There were also other researches in deep reinforcement learning leveraging causal relation (e.g., [i] and [ii] below). The title and the first contribution of this paper should be more specific and the related work should be included in the main text.
> There were unclear points in the implementation (without sharing codes) and the results (below).
> [i] W. Shi et al. Self-Supervised Discovering of Causal Features: Towards Interpretable Reinforcement Learning, arXiv, 2020 [ii] W. Agnew and P. Domingos. Relevance-Guided Modeling of Object Dynamics for Reinforcement Learning, arXiv, 2020”**
>
> A: The comments are addressed:
> 1. The title has been changed to "Deep Reinforcement Learning with Causality-based Intrinsic Reward" to be more specific on the contributions of the paper.
> 2. We add several discussions in section 3.1 (last paragraph) and section 5 of related work. More related work are presented (including two papers i and ii).  Please kindly refer to our uploaded revised version.
> 3. As we promised in the last two paragraph of the introduction, the environment and the algorithm are "to be released to the community". This sentence would definitely appear in our camera-ready version. In fact, we already get the official approval of this project to be open-sourced from our institution, and you can expect it to be released soon. For easy reference and reproducibility, we also provide many details in appendix.
>
> **Q: “There is little information about the algorithm of Relational DRL (Zambaldi et al., 2018) as one of the baseline. Relational DRL seems to be an attention-based A2C algorithm, but is the difference only related to the attention-based approach? At least, whether it is an A2C algorithm or not should be clarified.”**
>
> A: RDRL is actually an A2C algorithm, with extra attention based feature extration layers. To make this clarified, we add "We compare CARE with flat DRL method (A2C) and Relational DRL \citep{rdrl}, which is an A2C based algorithm and uses multi-head attention to extract relation between entities. " in the first paragraph of section 4.
>
> **Q: “In both experiments, is the number of categories K the number of agents? This was not specified.”**
>
> A: The K here refers to number of categories of entities, and this is in consistency with the definitions in section 3. To make this clear, we add specifications of K in section 4.1 and 4.2.
>
> **Q: “I did not understand Figure 5 right and the sentence (”The value of coordinate (x, y) in the heatmap corresponds to the effect of the agent on the ewe when it is on position (x, y) and taking action drive”). The correspondence of xy coordinates with Figure 5 left might help my understanding.”**
>
> A: Figure 5 is used to test if the effect of the agent's action on the entity is correctly estimated, which is for computing the intrinsic reward as showed by Eq 15. When the sheep are in positions as showed by figure 5 left, we place the agent in the positions coordinated by (x,y) and record the effect of the agent’s action on the ewe (showed by color). In other words, the value in coordination (x,y) records the effect of taking the action "drive" when the agent is at position (x,y). One can see that the closer the agent is to the ewe, the larger the effect would be. This is reasonable since the influence of the agent's action on ewe becomes larger if they are closer. To make this clear to readers, we make clarifications using the texts around the figure 5.
>
> Thank you again for your valuable comments that help a lot to improve the paper!

---

> > ### Comment · AnonReviewer3 · 2020-11-24
> > **About Figure 5**
> >
> > Thank you for the response. Most of the parts were clarified, but Figure 5 is still unclear.
> > I understood Figure 5 left (situation). In your response such that
> >
> > “we place the agent in the positions coordinated by (x,y) and record the effect of the agent’s action on the ewe (showed by color)”,
> >
> > where the agent was placed in Figure 5 right? Is the center of a colored square in Figure 5 right a white eve in Figure 5 left?  If the colored square in Figure 5 right presents the effect of the agent’s action on the ewe, the effect is strong except for the top row. Is it right?

---

> > > ### Author Response · Authors · 2020-11-24
> > > **About Figure 5**
> > >
> > > Thank you a lot for the reply.
> > >
> > > **Q: "About Figure 5, where the agent was placed in Figure 5 right?"**
> > >
> > > A: We place the agent on every (x, y) in the gridworld (from (0,0) to (17,17)). For each (x,y), we compute I_{ewe}(s,drive) and record it in a matrix R with size 18*18. For example, we place the agent at (0, 0), and comput I_{ewe}(s,drive). The result is the saved in R[0][0]. After getting all results like this, we visualize the matrix R as a heatmap.
> > >
> > > **Q: "Is the center of a colored square in Figure 5 right a white eve in Figure 5 left?"**
> > >
> > > A: Yes, the center of the colored square in Figure 5 right corresponds to the position of the white ewe. Except for the square field,  effect of the agent on the ewe when it is placed on other field computed by our method is almost zero. This experiment result shows that the effect of the agent on the ewe is strong only when it is in the square field, and this is in line with the  true game rule and our method can capture this.
> > >
> > > **Q: "If the colored square in Figure 5 right presents the effect of the agent’s action on the ewe, the effect is strong except for the top row. Is it right?"**
> > >
> > > A: Yes, compared with other positions in the square field, the effect  computed by our method for the top row is relatively small.

---

### Official Review · AnonReviewer4 · 2020-10-28
**Interesting method for deep reinforcement learning that learns causal structure of the environment and leverages it to assist policy learning.**

**Rating:** 6
**Confidence:** 3

**Review:**

In this paper, the authors propose a new deep reinforcement learning algorithm that learns the causal graph representation of the environment and leverages it to assist policy learning.

The authors should define a critical entity since they mentioned it many times in the article. Since the learned graph considers categories as nodes, I did not understand
the reference of critical entities as top-ranked nodes in the graph (Sec. Calculating ACE).

I would like also to see a proper definition of the entities and how the learned causal graph of entities impacts the performance of the proposed method.

There is confusion about the nodes in the causal graph, sometimes the authors mention them as entities:

Abstract: “critical entities of the causal graph”
Introduction:: ”The proposed algorithm learns a graph to encode the relation information between entities”.

And they change to consider nodes as categories of entities:

Method: “we calculate ACE values between categories of entities, which are used for constructing a causal graph G”
Sec.3.1: “we focus on the causal relation among categories of entities.”

For my review, I am considering nodes as categories of entities, given that the authors discuss a restriction on learning the category causal graph
rather than the entity causal graph to make their approach scalable. But this inconsistency and lack of definition lead to confusion.

My main concern about this paper is about the experimentation, specifically: baselines.

I consider that the requirement of no prior knowledge is an interesting and valuable contribution of this paper. However, in scenarios in which we have such information, I do not consider that we should ignore it. Thus, I consider it is an open research question the application of the proposed method in these scenarios.
In this sense, I consider that the approaches proposed in [1,2] should be considered as baselines in an experiment where CARE receives the causal graph and does not
“update" it during the sequential decision-making process. The categories of the given entity causal graph could be inferred to build the causal relation among categories
of entities as assumed as input to CARE.

I did not understand why CARE is not appropriate to Text-Adventure Game [3].

[1] Tingwu Wang, Renjie Liao, Jimmy Ba, and Sanja Fidler. Nervenet: Learning structured policy with graph neural networks. In Proceedings of the 6th International Conference on Learning Representations (ICLR), 2018.
[2] Varun Kumar Vijay, Abhinav Ganesh, Hanlin Tang, and Arjun Bansal. Generalization to novel objects using prior relational knowledge. arXiv preprint arXiv:1906.11315, 2019.
[3] Vinicius Zambaldi, David Raposo, Adam Santoro, Victor Bapst, Yujia Li, Igor Babuschkin, Karl Tuyls, David Reichert, Timothy Lillicrap, Edward Lockhart, et al. Deep reinforcement learning with relational inductive biases. In International Conference on Learning Representations, 2018.

References should point to the published work rather than the arxiv entries, when the former is available (e..g. Neural Network Attributions: A Causal Perspective, ICML’19).

Typo:

 - Sec.3.2: “of several deconvlutional …” -> “of several deconvolutional …”

---

> ### Author Response · Authors · 2020-11-17
> **Thank you for your valuable comments. Here are our responses.**
>
> **Q: “The authors should define a critical entity since they mentioned it many times in the article. Since the learned graph considers categories as nodes, I did not understand the reference of critical entities as top-ranked nodes in the graph (Sec. Calculating ACE).
> I would like also to see a proper definition of the entities and how the learned causal graph of entities impacts the performance of the proposed method.
> There is confusion about the nodes in the causal graph, sometimes the authors mention them as entities:
> Abstract: “critical entities of the causal graph” Introduction:: ”The proposed algorithm learns a graph to encode the relation information between entities”.
> And they change to consider nodes as categories of entities:
> Method: “we calculate ACE values between categories of entities, which are used for constructing a causal graph G” Sec.3.1: “we focus on the causal relation among categories of entities.””**
>
> A: Critical entities here refer to entities belonging to top ranked categories. To address the comments, we make several revisions:
> 1. The related statements in abstract and introduction are revised to be in consistency with other parts.
> 2. Section 3.1 is rewritten to introduce the category of entities, and a new definition 3 is added in section 3.2 to formalize the criticality of category of entities as their rankings in the casual ordering (top ranked nodes) required by our algorithm.
>
> **Q: “My main concern about this paper is about the experimentation, specifically: baselines. I consider that the requirement of no prior knowledge is an interesting and valuable contribution of this paper. However, in scenarios in which we have such information, I do not consider that we should ignore it. Thus, I consider it is an open research question the application of the proposed method in these scenarios. In this sense, I consider that the approaches proposed in [1,2] should be considered as baselines in an experiment where CARE receives the causal graph and does not “update" it during the sequential decision-making process. The categories of the given entity causal graph could be inferred to build the causal relation among categories of entities as assumed as input to CARE.”**
>
> A: We give a training schema targeting the cases where prior causal graph is not obtainable, which often happens, and consider the graph learning part a contribution of this paper. In our environment, the learned graphs are just the true ones, and providing the true graphs to it would generate the same performance curve for CARE. Since baselines such as A2C do not require a prior graph, to ensure fairness we do not compare them with the ones you list, given our current game settings. It is indeed interesting to consider the case where the underlying graph is available. We are going to maintain an open-source repository consisting of multiple game environments, and some of them may provide a choice that the graph is given or not. Then we would be able to test more baselines (e.g. [1,2]) to further promote this line of research.
>
> **Q: “I did not understand why CARE is not appropriate to Text-Adventure Game [3].”**
>
> A: The CARE introduced in the paper takes matrix signals as input and uses convolutional layers. The inference of causal graph is by minimizing the state prediction error. Thus, it cannot be directly adopted to texts. However, the framework of CARE is adoptable to text games, if one uses language processing techniques to identify several categories of entities from texts, and analyze their causal relations.
>
> **Q: “References should point to the published work rather than the arxiv entries, when the former is available (e..g. Neural Network Attributions: A Causal Perspective, ICML’19). Typo: Sec.3.2: “of several deconvlutional …” -> “of several deconvolutional …””**
>
> A: The comments have been addressed and please kindly refer to the revised version.
>
> Thank you again for your valuable comments that help a lot to improve the paper!

---

### Official Review · AnonReviewer1 · 2020-10-28
**A novel approach to learn causal relations and used them to accelerate the learning process of deep reinforcement learning**

**Rating:** 6
**Confidence:** 4

**Review:**

The authors proposed to learn causal relations to accelerate the
learning process in deep reinforcement learning. The proposed approach
first learns a graph to represent the causal structure of the
environment calculating the Average Causal Effect (ACE) between different
categories. It uses an intrinsic reward to encourage interacting with
the most relevant entities of the causal graph which accelerates the
learning process.

The proposed approach first need to create a factored state space,
given by the user. It is not clear how intuitive is this process and
if it can be performed automatically.

They first learn a model of the environment, predicting the next state of
each category of entities. For that they obtain data of the agent
interacting with the environment and use an encoder-decoder approach
to obtain k categories. Then predict the next state based on the
current state and an action, where the decoder predicts the next state
of each category.

It is not clear how to determine the number of categories? (how many
decoders?) If there is a single agent interacting with different
objects, a new category needs to be defined for each object?

It then evaluates the ACE between every pair of categories. Following
Chattopadhyay et al., they used the causal attribution method to infer
the causal relations between entities. The proposed approach then
constructs a DAG, where an edge between category "i" and "j" is placed
if ACE i->j > ACE j->i. The DAG represents a causal ordering where
categories with higher ACE value are placed before. Each category is
given an intrinsic reward, in addition to the extrinsic reward, based on
its order.

The approach was tested on two domains and compared against a flat DRL
(A2C) and Relational DRL with clearly better results.

The results should also include the time used to obtain the causal graphs.

Typos:
... analysis et, al (Ren et al. ...
... deconvlutional ...
... cause that the ewe ... => ... cause the ewe ...
... the the ...

Pros:
- The use of causal relations to accelerate the learning process
Cons:
- The approach requires manual segmentation of the state space into
categories
- Tested on two new domains and compared against a baseline an another
algorithm, so it is not completely clear the generality of the approach.

---

> ### Author Response · Authors · 2020-11-17
> **Thank you for your valuable comments. Here are our responses.**
>
> **Q: “The proposed approach first need to create a factored state space, given by the user. It is not clear how intuitive is this process and if it can be performed automatically.”**
>
> A: The factored space contains subspace each recording the states of entities belonging to one category. To construct such a space, we need to identify: 1. how many categories. 2. category of each entity. Given an environment with multiple entities, intuitively objects with the same shape, color or size might be within the same category. Domain knowledge like the background of the game also provides hints for this. One may consider an intuitive example that a game contains interacting balls with 3 colors: white, green and blue. Then the user can define the number of category to be 3 and conduct causal analysis. We thus think constructing the factored space (identifying number of category) is intuitive. In case where the category of the objects is not available as a prior, one may apply some segmentation approaches such as shape or color analysis to obtain the category of objects. For example, object detection and segmentation approaches can be applied in case that images are input and categories need to be inferred. We add discussions about this in  section 3.1. We think automatic construction of such factored space is achievable.
>
>
> **Q: “It is not clear how to determine the number of categories? (how many decoders?) If there is a single agent interacting with different objects, a new category needs to be defined for each object?”**
>
> A: For the first question, you may refer to the answer above. We here assume that the category of objects is fixed in the game. If new objects are generated (for example, in the ant game, a new ant is born), its category falls into the existed pool and we do not need to add a new decoder for it. We do not yet consider the case where the number of category of objects is changing.  But this is an interesting direction for future research.
>
>
> **Q: “The results should also include the time used to obtain the causal graphs.”**
>
> A: Table 3 recording the time is available in the revised version in section 4.2, with analysis. Please kindly refer to the revised version.
>
>
> **Q: “Typos: ... analysis et, al (Ren et al. ... ... deconvlutional ... ... cause that the ewe ... => ... cause the ewe ... ... the the ...”**
>
> A: The listed ones are corrected, and we also go through the paper for typo correction.
>
> Thank you again for your valuable comments that help a lot to improve the paper!

---

### Author Response · Authors · 2020-11-19
**Summary of changes**

We would like to give our sincere thanks to all reviewers for their really valuable comments. To address the comments, we make changes to our paper, which are listed below. Please kindly refer to our new uploaded version.
1.	The title is changed to be more specific about the contributions of the paper.
2.	In the abstract and introduction, statements confusing “entity” and “category of entity” are revised to be consistent.
3.	Section 3.1 is rewritten to discuss more about the environment with entities belonging to different categories, including some examples and discussions about the availability of such information.
4.	Definition 3 about “what are important categories” is added to section 3.2.
5.	Clarifications about the K and some baselines  are added in section 4.
6.	A new experimental result recording the running time is added in section 4 (table 3).
7.	More related works are discussed in section 5.
8.	Typos are corrected. The references are changed to refer to published ones rather than arxiv version when the former is available.

---

### Decision · Program_Chairs · 2021-01-07
**Final Decision**

**Decision:**

Reject

**Comment:**

The work proposed to learn causal structure of the environment and use the average causal effect of different categories of the environment, between the current and next state after performing an action as intrinsic reward to assist policy learning. While the reviewers find the ideas presented in the paper interesting and of potential, there are some concerns regarding proper introduction and comparison to related works, and clarity of the algorithm itself. While the two experimental results presented in the paper do show the potential of the work, it is missing an important baseline to disentangle the effectiveness of introducing the causal structure alone vs intrinsic reward. For example, how would A2C with curiosity or surprised based intrinsic reward, which also introduce the surprisingness of the next state as a result of performing an action as additional reward perform on these tasks?